# Early Diagnosis of Tumorigenesis via Ratiometric Carbon Dots with Deep-Red Emissive Fluorescence Based on NAD^+^ Dependence

**DOI:** 10.3390/molecules29225308

**Published:** 2024-11-11

**Authors:** Lan Cui, Weishuang Lou, Mengyao Sun, Xin Wei, Shuoye Yang, Lu Zhang, Lingbo Qu

**Affiliations:** 1College of Chemistry, Zhengzhou University, Zhengzhou 450001, China; 2College of Biological Engineering, Henan University of Technology, Zhengzhou 450001, China; 13598611016@163.com (W.L.); 17737879593@163.com (M.S.); 18236139462@163.com (X.W.); yangshuoyecpu@163.com (S.Y.); zhanglu@haut.edu.cn (L.Z.)

**Keywords:** ratiometric fluorescence, carbon dots, NAD^+^, tumorigenesis

## Abstract

The early diagnosis of tumorigenesis is crucial for clinical treatment, but the resolution and sensitivity of conventional short-wavelength biomarkers are not ideal because of the complicated interference in living tissue. Herein, a nicotinamide adenine dinucleotide (NAD^+^)-responsive probe with deep-red emissive ratiometric fluorescence was synthetized as a promising target for energy metabolism patterns during tumorigenesis. Interestingly, the solvents H_3_PO_4_ and 2,2′-dithiodibenzoic acid enhanced the red emission (640 and 680 nm) of o-phenylenediamine-based carbon dots (CDs), leading to the formation of a nanoscale graphite-like skeleton covered with -P=O, -CONH-, -COOH and -NH_2_ on their surfaces. Meanwhile, this method exhibited high sensitivity to the discriminating target NAD^+^, with a detection limit of 63 μM due to the inner filter effect and fluorescence resonance energy transfer process between NAD^+^ and CDs, which is superior to the reported capillary electrophoresis and liquid chromatographic detection methods (the reported detection limit was about 0.2 mM) in complex biological samples and even cancer cells. Encouragingly, NAD^+^ significantly promoted nucleus-targeting fluorescence and cell migration compared to GSH and pH stimulation, which were gradually eliminated in human hepatocellular carcinoma (HepG2) cells after 2-deoxy-d-Glucose inhibited the glycolytic phenotype. The proposed method holds great potential for the temporal and spatial resolution of NAD^+^-dependent tumor diagnosis in complex living systems.

## 1. Introduction

Aerobic glycolysis is an abnormal energy metabolism pattern that is closely associated with tumor occurrence and progression compared with the oxidative phosphorylation metabolism in normal cells [1]. Nicotinamide adenine dinucleotide (oxidized state—NAD^+^; reduced state—NADH) is the potential diagnostic and therapeutic target of aerobic glycolysis metabolism during tumorigenesis, especially as it is more highly expressed in hepatoma cells [2,3,4]. In addition, hepatocellular cancer is a serious disease with insidious incidence and a high mortality rate [5,6]. The identification of NAD^+^-dependent tumorigenesis plays a crucial role in better understanding the genesis and progression of tumors [7,8].

Fluorescence imaging for NAD^+^/NADH detection is superior in its simplicity and high sensitivity to electrochemistry, liquid chromatography and capillary electrophoresis, but its imaging depth and temporal and spatial resolution are limited in complicated living systems [9,10]. To date, carbon dots (CDs) have attracted tremendous attention in biomedical applications due to their chemical inertness, excellent photon and thermal stability and excellent biocompatibility [11,12,13]. The presence of the donor and recipient fluorescence resonance energy transfer process prompts the green fluorescence of 2,3-diaminophenothiazine-based CDs as a single imaging mode in spontaneous hepatocellular carcinoma [14]. Unfortunately, false signals often appear because the blue fluorescence background of living tissue interferes with the resolution of biological imaging [15,16]. How to avoid the interference of living tissues and the concentration limitation of a single peak has become key to fluorescence analysis in tumor diagnosis [17,18].

To overcome these drawbacks, deep-red (650–900 nm) fluorescent probes are gradually emerging as another promising direction due to their deep-tissue penetration and minimal interference with self-fluorescence [19,20]. In particular, the ratiometric probe based on the ratio of the fluorescence intensity of two peaks has great advantages in eliminating various external interferences such as photobleaching, false positive reactions, instrument errors, and small changes in the solution environment [21,22]. Moreover, its visual detection based on a wide range of color changes can be better realized and its detection accuracy can be improved through the inverse change in the double-response signal [23,24]. Thus, intrinsic red-green-emissive ratiometric probes are urgently required for discrimination of NAD^+^.

Herein, ratiometric probe CDs with deep-red emissive fluorescence were designed and synthesized to monitor NAD^+^ using 2,2′-dithiodibenzoic acid and o-phenylenediamine as carbon and nitrogen sources (Figure 1). There is a strong red shift from green to red fluorescence after they react with NAD^+^. Since the optimal excitation/emission wavelengths of NAD^+^ are 423/476 nm [14], the emission wavelength of the CDs shifted to a broad peak at 400–600 nm in the presence of NAD^+^, which suggests the inner filter effect (IFE) and a fluorescence resonance energy transfer (FRET) process between NAD^+^ and the CDs. Then, NAD^+^ can be determined with the ratio of fluorescence intensity at 640 nm to that at 520 nm (F_640_/F_520_); it exhibited an excellent linear relationship with a detection limit as low as 63 μM. However, its migration ability and red to green ratio of fluorescence intensity (F_Red_/F_Green_) were significantly reduced after the 2-deoxy-d-Glucose (2-DG)-induced suppression of the glycolytic phenotype in HepG2 cells. The dual-emissive ratiometric CDs reported here can provide a powerful and sensitive visualization strategy for preliminary diagnosis and a real-time detection platform in NAD^+^-dependent tumorigenesis.

## 2. Results and Discussion

### 2.1. Effect of Solvents and NAD^+^ on Absorption and Excitation–Emission Spectra of CDs

As shown in Figure 2A,B, solvents and 2,2′-dithiodibenzoic acid exhibited significant effects on absorption; the solvent H_3_PO_4_ was more inclined to prompt long-wavelength absorption on the o-phenylenediamine-based CDs. The absorption spectra of the CDs (Figure 2C,D) shows two absorption peaks, at 284 nm and 390 nm, due to the π-π* transitions in the aromatic sp^2^ domains (C=C, C-C). In addition, the dominating absorption bands at 566 nm and 615 nm were attributed to the n-π* transitions in multi-conjugated C=O/C=N and C-S, respectively [25].

Figure 2E,H show that 80% (volume fraction) H_3_PO_4_ increased the long-wavelength excitation and emission characteristics to 640 and 680 nm after 566 and 615 nm excitation, respectively, whereas the emission wavelength of 520 nm was relatively weaker under 390 nm excitation. Interestingly, the fluorescence emission intensity at 640 nm and 520 nm exhibited an opposite trend, increasing the concentration of NAD^+^. In addition, the emission wavelength of the CDs shifted to a wide peak at 400–600 nm, indicating an inner filter effect (IFE) and fluorescence resonance energy transfer (FRET) process between NAD^+^ and the CDs during the optimal excitation–emission wavelengths of NAD^+^ (423/476 nm) [14]. The fluorescence of CDs was observed to change from yellowish green to red under 365 nm UV light.

Herein, the ratio of fluorescence intensity at 640 nm to that at 520 nm (F_640_/F_520_) was investigated to evaluate the effect of NAD^+^ concentration on the CDs. As shown in Figure 2F, F_640_/F_520_ exhibited a good linear relationship with NAD^+^ when the concentration of NAD^+^ ([NAD^+^]) was maintained within the range of 0–40 mM. The linear equation was F_640_/F_520_ = 4.5421 + 0.2038 × [NAD^+^] (R^2^ = 0.994), and the limit of NAD^+^ detection (LOD) was as low as 63 μM in this ratiometric fluorescence system according to the calculation of 3δ/slope [26]. The fluorescence of the CDs was observed during irradiation under 365 nm ultraviolet light, which was gradually shifted from green to red fluorescence as the NAD^+^ concentration increased (Figure 2I).

It has been reported that the *QY*_R_ for RhB in ethanol solution is 0.68; the *QY*_R_ for quinine sulfate in 0.05 M sulfuric acid is 54.7%, and the refractive indices of the CDs and RhB/Quinine sulfate solution are 1.33 and 1.10, respectively. The calculated value for the *QY* of CDs is 11.6% according to Equation (1), which is close to the reported value [14].

The above results indicate that the ratiometric probe CDs with deep-red emissive fluorescence provide a highly selective and feasible method for the detection of NAD^+^ concentration in the tumor’s aerobic glycolytic metabolic pathway.

### 2.2. Characterization and Cytotoxicity Analysis of CDs

As shown in Figure 3A,B, the CDs exhibited well-dispersed and homogeneous particles with an average size of 8.7 nm.

Fourier transform infrared (FT-IR) spectra showed that the CDs had a broad peak at 3064–3400 cm^−1^, indicating the presence of O-H and N-H bonds on the surface of the CDs (Figure 3C). The characteristic absorption peak at 2973 cm^−1^ was attributed to the C-H stretching vibration, the absorption peak at 2480 cm^−1^ was ascribed to the P-OH bending vibration [27], the absorption peaks at 1682 and 1261 cm^−1^ due to the C=O and C-N stretching vibration, and the absorption peaks at 685 and 491 cm^−1^ were associated with the C-S and S-S vibration peaks [28,29].

The full X-ray photoelectron spectroscopy (XPS) spectra of the CDs contained typical peaks at 284, 400, 532, 164 and 133 eV that corresponded to C 1s, N 1s, O 1s, S 2p and P 2p [27], and their atomic ratios were calculated to 60.07%, 3.25%, 27.57%, 4.57% and 4.54%, respectively (Figure 3D). In Figure 3E, the high-resolution XPS spectrum of the C 1s band was separated into three peaks at 284.8, 286.2, 287.9 and 288.9 eV, which corresponded C-C/C=C, C-N/C-S/C-P=O, C=O and -COO-, respectively. The N 1s band contains two peaks at 399.4 and 400.9 eV, which were assigned to pyridinic C_3_-N and pyrrolic C_2_-N-H groups (Figure 3F). The S 2p band in Figure 3G was separated into three peaks at 163.35, 163.81 and 164.57 eV, which revealed the presence of S-C, S-H and S-S groups, respectively [30].

Based on the aforementioned characterizations, the reaction mechanism of the CDs could be deduced. CDs are a nanoscale graphite-like skeleton with defects that are caused by pyridinic nitrogen atoms and disulfide bonds and are covered with -P=O, -CONH-, -COOH and -NH_2_ on the surface of their symmetrical heterocycle rotatable structures.

As shown in Figure 3H,I, the hemolysis rate was lower than 5% when the concentration of CDs ranged from 0 to 500 μg/mL. The intracellular viability in HepG2 cells slightly decreased with the increase in the CD concentration and the extension of the incubation time. In addition, the viability was still above 80% after 24 and 48 h. The above results indicate that CDs exhibit good biocompatibility and lower toxicity.

### 2.3. Effects of NAD^+^ Concentration and Incubation Time on Intracellular Uptake

NAD^+^ is the potential diagnostic target of tumor proliferation, migration and malignant development on account of the abnormal glycolysis metabolism of tumor cells. 2-deoxy-d-Glucose (2-DG) is a glycolytic blocking agent that effectively interferes with the NAD^+^ level. As shown in Figure 4A, the concentration of NAD^+^ in HepG2 cells was significantly higher than that in normal tissues. In addition, the NAD^+^ concentration and NAD^+^/NADH ratio in HepG2 cells were significantly reduced after the glycolysis inhibition of 2-DG. Therefore, HepG2 cells were subsequently selected as the research object.

As shown in Figure 4B, the detection specificity of the CDs was studied, including for ions (100 μM of K^+^, Na^+^, Ca^2+^, Mg^2+^, Zn^2+^, Al^3+^, Fe^3+^), amino acids (like 100 μM of arginine (Arg), cysteine (Cys), tyrosine (Tyr)) and proteins and enzymes (like 100 U/L of human serum albumin (HSA), bovine serum albumin (BSA) and 50 μM glucose (Glu)). The F_Red_/F_Green_ value changed weakly, only exhibiting significant changes in the presence of NAD^+^. The above interference indicates that the sensor platform has excellent selectivity for NAD^+^.

The absorbance of CDs in HepG2 cells at 390 nm and 566 nm was measured by a multimode plate reader (Figure 4C). The Abs_390nm_ and Abs_566nm_/Abs_390nm_ showed a trend of rapid increase at first and then a slow increase while increasing the NAD^+^ concentration ([NAD^+^]), which was consistent with the result observed with the confocal laser scanning microscope. F_566nm_/F_390nm_ = −0.787 × e^−[NAD+]/23.55^ + 1.688, R^2^ = 0.9997. Moreover, the Abs_390nm_ and Abs_566nm_/Abs_390nm_ were reduced after the addition of glycolytic blocking agent 2-DG (Figure 4D).

As shown in Figure 5, the red luminescent region of CDs was mainly located in the nucleus and nuclear periphery of HepG2 cells, indicating that CDs could enter the cell interior through the cell membrane and nuclear membrane. The temporal comparison (Figure 5A) shows that CDs accumulated rapidly within 1–3 h and exhibited strong fluorescence intensity at 3 h, but gradually diminished at 7–48 h, indicating that CDs were gradually excluded from the cells after 3 h. Moreover, the red fluorescence of CDs was enhanced, while their green fluorescence was slightly weakened after the addition of NAD^+^, which was basically consistent with the trend of fluorescence intensity in vitro. The red fluorescence intensity gradually increased with the increase in the NAD^+^ concentration from 0 to 20 mM, but the changes were relatively slow after a pretreatment of NAD^+^ at 20 to 40 mM, demonstrating that the ideal concentration of NAD^+^ is 20 mM (Figure 5B).

The fluorescence intensity of the CDs was quantified by Image J software (Figure 5C,D). The ratio of red to green fluorescence intensity (F_Red_/F_Green_) exhibited time- and dose-dependence on NAD^+^ concentrations. The red fluorescence and F_Red_/F_Green_ ratio were also higher at first and then decreased over time after adding NAD^+^. The F_Red_/F_Green_ curve also showed a trend of a rapid rise and then a slow rise with the increase in the NAD^+^ concentration. The F_Red_/F_Green_ ratio showed an exponential relationship with the NAD^+^ concentration, F_Red_/F_Green_ = −0.434 × e^−[NAD+]/1.111^ + 0.991, R^2^ = 0.991. The detection limit was calculated to be as low as 69 μM, indicating a superior higher sensitivity than that of previously reported NAD^+^ methods in complex biological samples and even cancer cells (Table 1) [31,32,33,34,35].

The above results showed that the obtained CDs could rapidly detect intracellular NAD^+^ due to the inner filter effect (IFE) and fluorescence resonance energy transfer (FRET) process between NAD^+^ and CDs [14]. CDs could serve as fluorescence-sensing platforms for the early warning of tumor formation based on the visual detection of NAD^+^.

### 2.4. Migratory Capacity and Intracellular Uptake of CDs After Aerobic Glycolysis Inhibition

Figure 6 and Figure 7 show the effect of glycolysis inhibitor 2-deoxy-d-Glucose (2-DG) on the migratory capacity and intracellular uptake of CDs in HepG2 cells. As shown in Figure 6A,B, the tumor microenvironment, which as weakly acidic, redox-promoting (GSH), and the glycolytic metabolic initiator—nicotinamide adenine dinucleotide (NAD^+^)—promoted tumor migration. The migration rate increased from 3.82% (pH 7.4) to 11.96% (pH 6.5 + 10 mM GSH + 20 mM NAD^+^). In particular, NAD^+^ had a promotional effect on migration > GSH > pH. However, the migration distance of HepG2 cells to the scratch area was reduced by 2-DG to some extent [36]. Under the condition of pH 6.5 + 10 mM GSH + 20 mM NAD^+^, the mobility rate decreased from 11.96% to 8.89% after adding 2-DG (Figure 7A). These results indicated that cell growth could be significantly promoted by the tumor microenvironment, the weakly acidic reducing agent GSH and the glycolytic metabolism initiator NAD^+^, but glycolytic metabolism and cell migration could be significantly disrupted by the glycolysis inhibitor 2-DG.

Figure 6C,D and Figure 7B show the intracellular uptake analysis of carbon dots in HepG2 cells after the glycolysis inhibition of 2-deoxy-d-Glucose (2-DG). As shown in Figure 6C, without the pretreatment of 2-DG, the weakly acidic microenvironment, redox agent (GSH) and nicotinamide adenine dinucleotide (NAD^+^) could promote the intracellular uptake of carbon dots in HepG2 cells. The ratio of intracellular red to green fluorescence (F_Red_/F_Green_) increased from 0.67 (pH 7.4) to 0.99 (pH 6.5 + 10 mM GSH + 20 mM NAD^+^, Figure 7B), where the migration effect was NAD^+^ > GSH > pH. However, the intracellular red fluorescence intensity in each experimental group was reduced after the addition of 2-DG (Figure 6D), and F_Red_/F_Green_ decreased from 0.99 to 0.61, especially under pH 6.5 + 10 mM GSH + 20 mM NAD^+^ conditions (Figure 7B).

The above results indicate that the weakly acidic reducing agent GSH and glycolytic metabolism initiator NAD^+^ significantly promote the intracellular uptake of carbon dots. With the addition of 2-DG, the green fluorescence intensity increased slightly, but the red fluorescence intensity decreased significantly due to the 2-DG-induced destruction of glycolytic metabolism and regulation of intracellular NAD^+^ [37].

## 3. Methods and Methods

### 3.1. Materials

2,2′-dithiodibenzoic acid (DTSA), o-phenylenediamine (OPD), ethanol, acetic acid, phosphoric acid and N, N-Dimethylformamide were purchased from Macklin Biochemical Co., Ltd. (Shanghai, China). Nicotinamide adenine dinucleotide (NAD^+^) and 2-deoxy-d-Glucose (2-DG) were purchased from Sigma-Aldrich Co., Ltd. (Shanghai, China). Hoechst 33342 staining solution and reductive glutathione (GSH) were purchased from Solaibao Technology Co., Ltd. (Beijing, China). Human hepatocellular carcinoma (HepG2) cells were purchased from the Chinese Academy of Sciences (Beijing, China).

### 3.2. Instruments and Measurements

UV absorption was measured using a UV-2550 visible spectrophotometer (Shimadzu Corporation Ltd., Kyoto, Japan). The size diameter was estimated by a zeta-sizer nano ZS90 (Malvern Ltd., Leamington Spa, UK). The chemical structures were measured by a Fourier transform infrared spectrometer (Nicolet iS20, Thermo Fisher Scientific Ltd., Waltham, Massachusetts, USA) and X-ray photoelectron spectroscopy (XPS, Kratos AXIS Ultra DLD, Kratos Analytical Ltd., Manchester, UK). The morphologies were visualized by a transmission electron microscope (HT7700, HITACHI, Hitachi Ltd., Tokyo, Japan). Fluorescence spectra were recorded by a fluorescence spectrophotometer (Hitachi F-4600, Hitachi High-Technologies Corporation Ltd., Tokyo, Japan). The absorbance of each well was measured by a hybrid multi-mode microplate reader (Tacan Spark, Tecan Trading AG Ltd., Männedorf, Switzerland). The fluorescence intensity in each cell was recorded by a cell imaging analysis system (Axio Observer 3, Carl Zeiss AG Ltd., Oberkochen, Germany).

### 3.3. Preparation of CDs

CDs were prepared using a hydrothermal method. DTSA (0.009 g) and OPD (0.003 g) were dissolved in 3 mL of different solvents, including water, ethanol, acetic acid, phosphoric acid and N, N-Dimethylformamide. The above solutions were transferred to a polytetrafluoroethylene autoclave and reacted at 180 °C for 12 h, and then naturally cooled to 25 °C. The collected samples were separated and purified by centrifugation and dialyzed with a dialysis bag (with a molecular weight cut-off of 500 Da), respectively. Finally, the product was obtained by freeze-drying.

The quantum yield (*QY*) of CDs was calculated according to Equation (1):(1)$$QY={QY}_{R}\times \frac{F}{F_{R}}\times \frac{A_{R}}{A}\times \frac{\eta^{2}}{\eta_{R}^{2}}$$

*F*: the measurement of integrated emission intensity. *A*: the optical density. *η*: the refractive index of the solvent.

### 3.4. NAD^+^ Detection by CDs In Vitro

The response of CDs (2 mg/mL) to NAD^+^ (2 mL, 0–40 mM) was measured by UV-vis absorption and fluorescence detection. Subsequently, the excitation–emission wavelengths and fluorescence intensities at 390, 566 and 615 nm were measured by fluorescence spectrometry.

### 3.5. Hemolytic Analysis of CDs

Mouse blood was placed in a glass tube and then EDTA-2Na was immediately added to prevent coagulation. Defibrinated blood was obtained after stirring to remove the fibrin. Then, red blood cells (RBCs) were precipitated after the addition of saline and centrifugation until the supernatant was free of red color. Subsequently, RBCs were prepared as a 2% cell suspension in saline for hemolytic analysis and stored at 4 °C.

The following were added 0.5 mL of 2% RBC suspension: 0.5 mL of saline (negative control, 0% hemolysis), 1% TritonX-100 (positive control, 100% hemolysis) and different concentrations of carbon dots. After 2 h of incubation, samples were collected after centrifugation and then placed into a 96-well plate to measure their absorbance at 540 nm. The hemolysis rate was calculated according to Equation (2).
(2)$$Hemolytic\;Rate\;\left(\%\right)=\frac{{Abs}_{sample}-{Abs}_{negative\;control}}{{Abs}_{positive\;control}{-Abs}_{negative\;control}}\times 100\%$$

*Abs_sample_*: the absorbance of the experimental group; *Abs_negative control_*: the absorbance of the negative control; *Abs_positive control_*: the absorbance of the positive control.

### 3.6. Cytotoxicity Assays of CDs

HepG2 cells were inoculated in 96-well plates with 10^4^ cells/well at 37 °C for 24 h. Then, CDs were added to 96-well plates at concentrations of 1, 10, 100, 250 and 500 μg/mL. After 24 and 48 h incubation periods, the cytotoxicity was measured by the MTT method and according to Equation (3).
(3)$$Cell\;viability\;(\%)=\frac{{Abs}_{sample}-{Abs}_{control}}{{Abs}_{cell}{-Abs}_{control}}\times 100\%\;\;\;\;\;\;\;\;$$

*Abs_sample_*: the absorbance of cells after incubation with nanomedicine; *Abs_cell_*: the absorbance of cells without nanomedicine treatment; *Abs_control_*: the absorbance of culture medium at 490 nm.

### 3.7. Cellular Uptake of CDs

HepG2 cells were inoculated in 6-well plates at a density of 10^5^ cells/well. Then, CDs (75 μg/mL) were added and incubated for 1, 3, 7, 12, 24 and 48 h. The nucleus was stained by Hoechst 33342 and observed with a confocal laser scanning microscope.

The effect of NAD^+^ concentration on the fluorescence intensity was investigated after the co-culturation of NAD^+^ for 3 h.

### 3.8. Migration Assay

HepG2 cells were seeded into 6-well plates at a density of 10^5^ cell/well. After 80% cell coverage, mediums containing 5 mM of glycolysis inhibitor 2-deoxy-d-Glucose (2-DG) were used to pretreat the experimental group for 5 h [38,39]. Then, confluent cells were scratched with a 200 μL micropipette tip and washed with PBS to remove all the cell debris of the wounded layers. Subsequently, culture mediums containing carbon dots (fluorescent carbon dots were prepared to a volume fraction of 80% H_3_PO_4_, 75 μg/mL) under different pH, GSH and NAD^+^ conditions (pH 7.4, pH 7.4 + 10 mM GSH, pH 7.4 + 20 mM NAD^+^, pH 6.5, pH 6.5 + 10 mM GSH, pH 6.5 + 10 mM GSH + 20 mM NAD^+^) were then added and incubated for 7 h. Images of wound healing were taken at 0 and 7 h under an inverted light microscope at 20× magnification. The wounding area was quantified by ImageJ 1.47v/Java 1.6.0_20 software.

### 3.9. Effect of NAD^+^ on the Cellular Uptake of CDs During Glycolysis Inhibition

HepG2 cells were seeded into 6-well plates at a density of 10^5^ cell/well. After 80% cell coverage, mediums containing 5 mM glycolysis inhibitor 2-deoxy-d-Glucose (2-DG) were used to pretreat the experimental group for 5 h. Subsequently, culture mediums containing carbon dots were added and incubated under different pH, GSH and NAD^+^ conditions for 7 h. The internalization of CDs was observed with a confocal laser scanning microscope. The fluorescence intensity was quantified by ImageJ 1.47v/Java 1.6.0_20 software.

### 3.10. Statistical Analysis

The experimental data were presented as the mean ± standard deviation of at least three measurements. Student’s *t*-test and a one-way ANOVA statistical analysis were performed using program for social sciences (SPSS 25.0) software. The results were considered statistically significant and extremely statistically significant at a level of *p* < 0.05, *p* < 0.01 and *p* < 0.001, respectively.

## 4. Conclusions

In this study, NAD^+^-dependent ratiometric CDs with deep-red emissive fluorescence were rationally designed through a solvent thermal reaction, using 2,2′-dithiodibenzoic acid and o-phenylenediamine as carbon and nitrogen sources, and 80% (volume fraction) H_3_PO_4_ as the solvent. The CDs achieved a high sensitivity to NAD^+^, with a detection limit of 63 μM through quantitative fluorescence intensity analysis, which is superior to that of reported NAD^+^ assays. Importantly, the proposed CDs can successfully monitor NAD^+^ fluctuation under metabolic perturbation, significantly promoting nucleus-targeting in the NAD^+^-tumor microenvironment compared to weakly acidic and reducing GSH stimulation. Furthermore, this strength was significantly eliminated in complex biological samples and even in cancer cells, especially in HepG2 cells after 2-DG-induced glycolytic phenotype suppression. Above all, ratiometric CDs with deep-red emissive fluorescence, in this proposed method, greatly facilitate the accurate and sensitive detection of NAD^+^ in a biomedical laboratory, holding significant potential in NAD^+^-related disease diagnosis and clinical research.

## Figures and Tables

**Figure 1 molecules-29-05308-f001:**
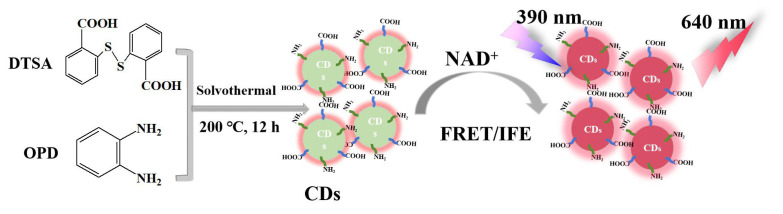
Schematic diagram of ratiometric probe CDs with deep-red emissive fluorescence for NAD^+^ determination.

**Figure 2 molecules-29-05308-f002:**
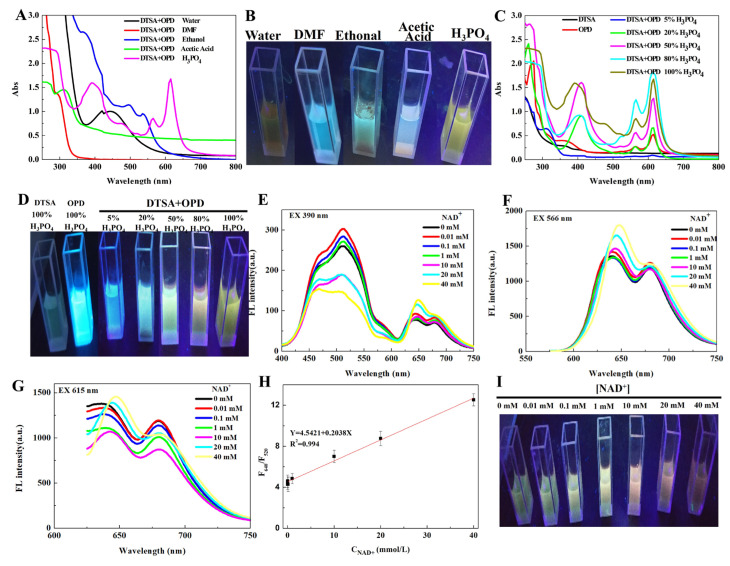
Absorption and excitation–emission characteristics of CDs. (**A**–**D**) UV-vis absorption spectrum under different solvent and H_3_PO_4_ concentrations ((**B**,**D**) are photos of CDs under 365 nm UV light). (**E**–**G**) The effect of NAD^+^ concentration on the excitation–emission wavelength of CDs at excitation wavelengths of 390, 566 and 615 nm. (**H**) Fitting curve of fluorescence intensity of ratiometric CDs. (**I**) Photos of CDs under 365 nm UV light.

**Figure 3 molecules-29-05308-f003:**
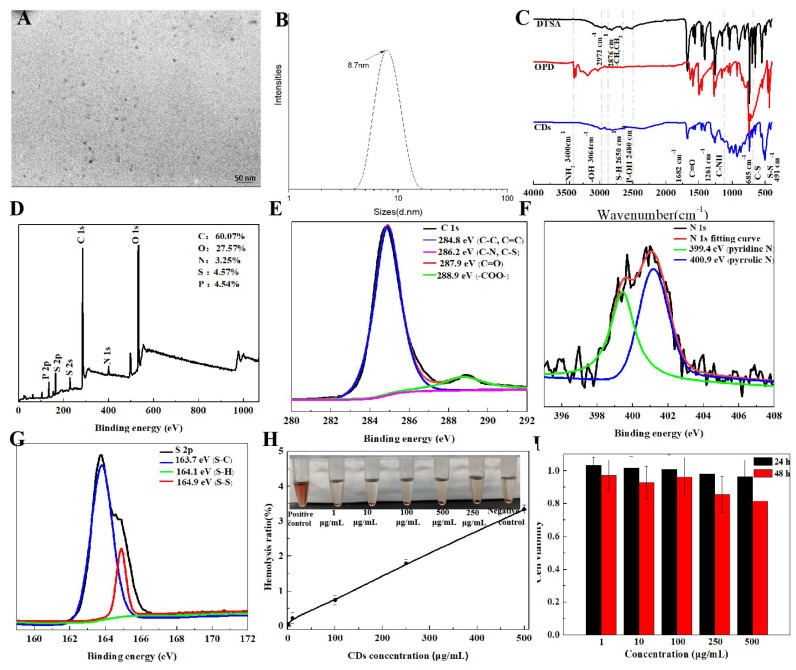
Characterization and cytotoxicity analysis of CDs. (**A**) TEM image. (**B**) Particle size. (**C**) FT-IR spectra. (**D**) XPS spectrum. (**E**–**G**) C 1s, N 1s and S 2p spectra. (**H**) Hemolysis rate analysis (inset, determination of hemolysis). (**I**) Cytotoxicity analysis.

**Figure 4 molecules-29-05308-f004:**
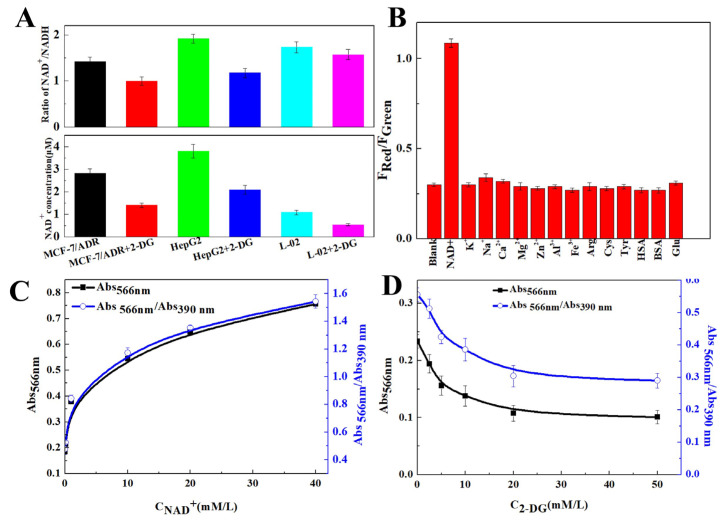
Effects of NAD^+^ and 2-DG levels on the intracellular uptake of CDs. (**A**) The intracellular NAD^+^ levels and NAD^+^/NADH ratio of MCF-7/ADR, HepG2 and L-02 cells. (**B**) Effects of common interfering agents on NAD^+^ detection. (**C**,**D**) Absorbance of CDs in HepG2 cells after NAD^+^ and 2-DG treatment.

**Figure 5 molecules-29-05308-f005:**
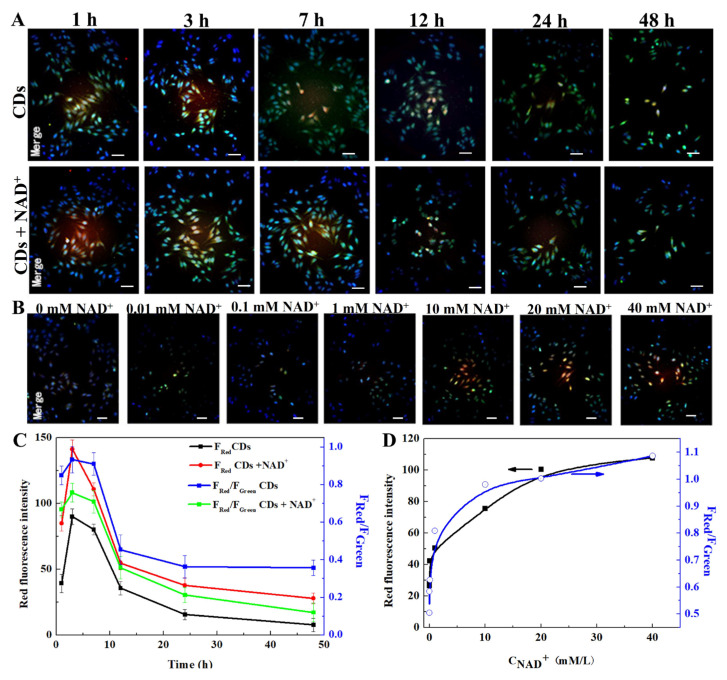
Fluorescence intensity analysis of CDs in HepG2 cells under different conditions. (**A**) Fluorescence images of CDs in the absence or presence of NAD^+^ at 1, 3, 7, 12, 24 and 48 h. (**B**) Fluorescence images after the pretreatment of NAD^+^. (**C**,**D**) Red fluorescence intensity and F_Red_/F_Green_ analysis at different incubation times and NAD^+^ concentrations.

**Figure 6 molecules-29-05308-f006:**
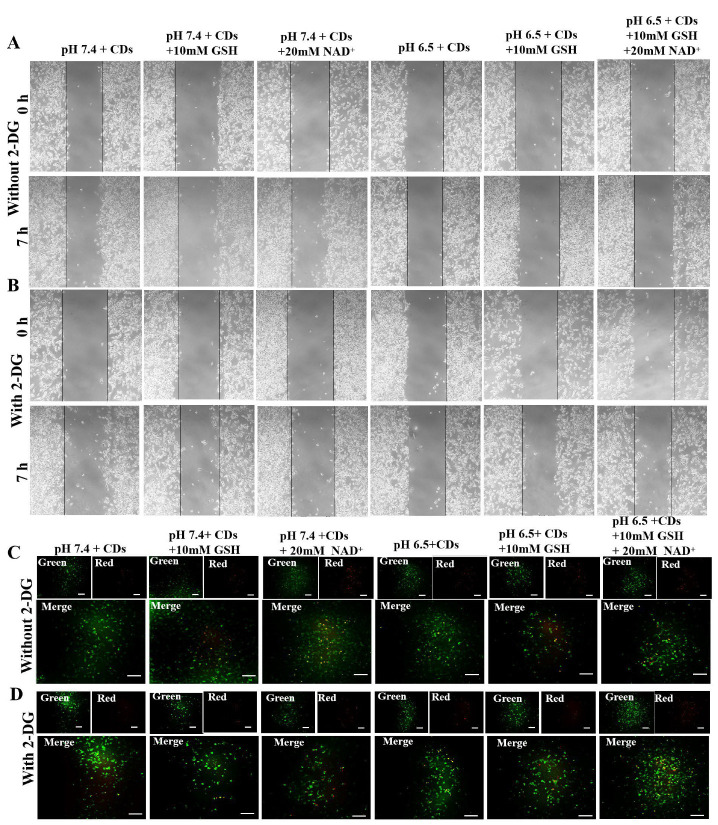
Effect of glycolysis inhibition on the intracellular uptake of CDs in HepG2 cells. (**A**,**B**) Cell migration at 0 and 7 h without or with the glycolysis inhibition of 2-DG. (**C**,**D**) Intracellular uptake of CDs without or with 2-DG pretreatment for 5 h (scale bar: 20 μm).

**Figure 7 molecules-29-05308-f007:**
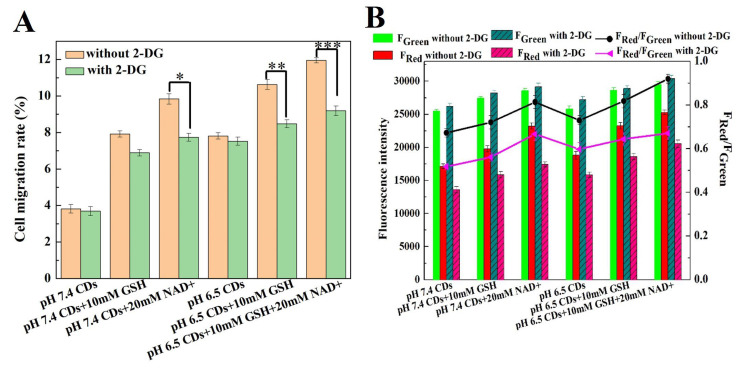
Cell migration and intracellular uptake of CDs in HepG2 cells after glycolysis inhibition of 2-DG. (**A**) Cell migration analysis at 0 and 7 h without or with 2-DG pretreatment. (**B**) Intracellular fluorescence intensity without or with 2-DG pretreatment for 5 h. *, ** and *** were considered as statistical significance and extremely statistical significance at a level of *p* < 0.05, *p* < 0.01 and *p* < 0.001, respectively.

**Table 1 molecules-29-05308-t001:** Comparison of the proposed method with the reported fluorescent methods for NAD^+^ assays.

Strategy	LOD	Linear Range	Real Sample Type	Ref.
Capillary electrophoresis	0.2 mmol/L	0.1–2 mmol/L	Rat heart myoblasts	[31]
Liquid chromatographic detection methods	6 μmol/L	5–100 μmol/L	Human astroglioma cells	[32]
Electrochemical sensors	22.3 μmol/L	10–100 μmol/L	-	[33]
UV	0.1 μmol/L	0–1 μmol/L	-	[34]
Enzyme colorimetry	-	-	-	[35]

## Data Availability

The datasets used and/or analyzed during the current study are available from the corresponding author upon reasonable request.

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
