# Peer review of "Early Diagnosis of Tumorigenesis via Ratiometric Carbon Dots with Deep-Red Emissive Fluorescence Based on NAD+ Dependence"

_molecules, 2024, doi:10.3390/molecules29225308_

Round 1
Reviewer 1 Report
Comments and Suggestions for Authors
The manuscript addresses an interesting subject with implications for the early diagnosis and treatment of tumours.
The authors are recommended to consider the following comments:
- In the introduction, describe other similar results obtained with the same system;
- increase the clarity and resolution of figures 2, 3 and 7 (or at least of the legends and insets);
- figure 6, include in the legend what merge stands for;
-consider all equations and re-number them in the order they appear in the text. It is hard to follow equations that are inserted in the body text;
- besides the presentation of the results shown in the figures, compare them with similar published results;
- include the producer for all of the used equipment in the experimental part;
- indicate how many repetitions for one measurement;
- correlate the annotation in definitions of the variables in equation (1) with the ones in the equation;
- include/extend the future perspectives in the concluding section.
Comments on the Quality of English LanguageModerate English language improvement.
Author Response
Response to Reviewer 1
We gratefully appreciate your valuable comments and constructive suggestions for improving the quality of this manuscript. The following are our point-by-point responses.
Comments 1: In the introduction, describe other similar results obtained with the same system.
Response 1: Other similar results obtained with the same system was added in the introduction. The conventional short wavelength biomarker often suffers from unsatisfactory resolution and sensitivity in virtue of the complicated interference of living tissue. Meanwhile, this method exhibited high sensitivity to discriminating target NAD+ with a detection limit of 63 μM due to the inner filter effect and fluorescence resonance energy transfer process between NAD+ and CDs, which exhibited high sensitivity in complex biological samples and even in cancer cells, which is superior to the reported NAD+ assays.
Comments 2: increase the clarity and resolution of figures 2, 3 and 7 (or at least of the legends and insets).
Response 2: The clarity and resolution, and the legends and insets were respectively increased in figures 2, 3 and 7.
Comments 3: figure 6, include in the legend what merge stands for.
Response 3: The merge legend in figure 6 stands for the merge of red and green fluorescence of CDs in HepG2 cells observed by confocal laser scanning microscope.
Comments 4: consider all equations and re-number them in the order they appear in the text. It is hard to follow equations that are inserted in the body text.
Response 4: All equations are re-numbered in the order that appeared in the text of Materials and Methods. The three linear equations were the results of fitting experimental data (The fluorescence invitro was recorded by fluorescence spectrophotometer, the absorbance of each well was measured by Hybrid Multi-mode Microplate Reader, the fluorescence intensity of in each cell was recorded by Cell imaging analysis system) recorded by fluorescence spectrophotometer through Origin software.
Comments 5: besides the presentation of the results shown in the figures, compare them with similar published results.
Response 5: Similar published results were sorted and quoted in Table 1 and highlighted in the manuscript. The detection limit was calculated as low as 69 μM with high sensitivity in complex biological samples and even in cancer cells, which is superior to the reported NAD+ assays (Table 1).
Table1 Comparison of the proposed method with the reported fluorescent methods for NAD+ assay.
Strategy |
LOD |
Linear range |
Real sample type |
Ref. |
Capillary electrophoresis |
0.2 mmol/L |
0.1–2 mmol/L |
Rat heart myoblasts |
31 |
Liquid chromatographic detection methods |
6 μmol/L |
5–100 μmol/L |
Human astroglioma cells |
32 |
Electrochemical sensors |
22.3 μmol/L |
10–100 μmol/L |
– |
33 |
UV |
0.1μmol/L |
0–1μmol/L |
– |
34 |
Enzyme colorimetry |
– |
– |
– |
35 |
Comments 6: include the producer for all of the used equipment in the experimental part.
Response 6: All of the producers of the used equipment were added in the experimental part.
Comments 7: indicate how many repetitions for one measurement.
Response 7: The experimental data were presented as the mean ± standard deviation from at least three measurements and highlighted in the manuscript.
Comments 8: correlate the annotation in definitions of the variables in equation (1) with the ones in the equation
Response 8: The annotation in definitions of the variables in equation (1) was modified and correlated with the ones in the equation.
Comments 9: include/extend the future perspectives in the concluding section.
Response 9: The future perspectives in the concluding section was extended and highlighted in the manuscript.
Thank you very much again for the carefully reviews and valuable comments.
Best regards,
Lan Cui and Lingbo Qu,
College of Chemistry, Zhengzhou University, Zhengzhou 450001, PR China; College of Biological Engineering, Henan University of Technology, Zhengzhou 450001, PR China.

Reviewer 2 Report
Comments and Suggestions for Authors
The research is focused on the preparation and testing of fluorescent ratio-metric carbon QD for tumor cells visualization. In my opinion the topic is original and relevant to the field. To the best of my knowledge, the original approach based on the application of ratio-metric carbon QDs with deep red or IR emission for this task is quite novel.
The authors mainly address the problem of avoiding of strong interference in living tissues while detecting florescence signal from tumor cells by using as a signal a spectral ratio instead of a single-wavelength fluorescence intensity.
The methodology of the research is generally correct. The limit of detection and the linearity of florescence response were tested for model solutions NAD+ and HepG2 cells in the presence of multiple potential interfering substances. This part of the research is undoubtedly original and could be considered as a significant contribution to the field.
The conclusions made by the authors are supported by the evidence and arguments presented.
The paper is well written and well structured, the list of references is adequate to the current state of this field of research; all references are appropriate.
There are several minor issues that should be corrected:
1) Fitting curves of fluorescence intensity of ratio-metric CDs (Fig. 2F and Fig. 5D) do not clearly demonstrate the linearity of the curves for small concentrations of NAD+ near the LOD.
2) The authors clearly indicate the CDs as near-infrared while all emission peaks are located in the far red spectral region below 800 nm.
Author Response
Response to Reviewer 2
Dear reviewer 2,
We gratefully appreciate your valuable comments and constructive suggestions for improving the quality of this manuscript. The following are our point-by-point responses.
Comments 1: Fitting curves of fluorescence intensity of ratio-metric CDs (Fig. 2F and Fig. 5D) do not clearly demonstrate the linearity of the curves for small concentrations of NAD+ near the LOD.
Response 1: The linear equation is F640/F520 = 4.5421 + 0.2038×[NAD+] (R2 = 0.994), as The concentration of NAD+ ([NAD+]) was maintained in the range of 0–40 mM (namely, 0–40000 μM). The limit of NAD+ detection (LOD) was as low as 63 μM. The above results could demonstrate the linearity of the curves for small concentrations of NAD+ near the LOD.
Comments 2: The authors clearly indicate the CDs as near-infrared while all emission peaks are located in the far red spectral region below 800 nm.
Response 2: The solvents H3PO4 and 2,2'-dithiodibenzoic acid enhanced the red emission (640 and 680 nm) of o-phenylenediamine based carbon dots (CDs), the “near-infrared” was change by “Deep-Red Emissive Fluorescence”.
Thank you very much again for the careful reviews and valuable comments.
Best regards,
Lan Cui and Lingbo Qu,
College of Chemistry, Zhengzhou University, Zhengzhou 450001, PR China; College of Biological Engineering, Henan University of Technology, Zhengzhou 450001, PR China.

Round 2
Reviewer 1 Report
Comments and Suggestions for Authors
All recommendations were addressed, except for the ”In the introduction, describe other similar results obtained with the same system.”
The authors included the text in the abstract, while the recommended section does not consider a comparison with similar results.
Comments on the Quality of English Language
Minor editing of English language is necessary.
Author Response
Response to Reviewer
We gratefully appreciate your valuable comments and constructive suggestions for improving the quality of this manuscript. The following are our point-by-point responses.
Comments 1: All recommendations were addressed, except for the ”In the introduction, describe other similar results obtained with the same system.”
Response 1: The other similar results were obtained with the same system in the introduction and the results and discussion, these were highlighted in the manuscript. “Fluorescence imaging for NAD+/NADH detection is superior in simplicity and high sensitivity than electrochemistry, liquid chromatography and capillary electrophoresis, but its imaging depth, temporal and spatial resolution are limited in complicated living systems in vivo9, 10.” The detection limit was calculated as low as 69 μM, indicating a superior higher sensitivity than the previously reported NAD+ methods (the reported detection limit was about 0.2 mM) in complex biological samples and even cancer cells.
Comments 2: The authors included the text in the abstract, while the recommended section does not consider a comparison with similar results.
Response 2: this method exhibited high sensitivity to discriminating target NAD+ with a detection limit of 63 μM due to the inner filter effect and fluorescence resonance energy transfer process between NAD+ and CDs, which is superior to the reported capillary electrophoresis and liquid chromatographic detection methods (the reported detection limit was about 0.2 mM) in the complex biological samples and even cancer cells.
Comments 2: Minor editing of English language is necessary.
Response 3: The English language was further improved and highlighted in the manuscript.
Thank you very much again for the careful reviews and valuable comments.
Best regards,
Lan Cui and Lingbo Qu,
College of Chemistry, Zhengzhou University, Zhengzhou 450001, PR China; College of Biological Engineering, Henan University of Technology, Zhengzhou 450001, PR China.
